# Convention on Biological Diversity (CBD) and the Nagoya Protocol: Implications and Compliance Strategies for the Global Coffee Community

**DOI:** 10.3390/foods13020254

**Published:** 2024-01-13

**Authors:** Dirk W. Lachenmeier, Christophe Montagnon

**Affiliations:** 1Chemisches und Veterinäruntersuchungsamt (CVUA) Karlsruhe, Weissenburger Strasse 3, 76187 Karlsruhe, Germany; 2RD2 Vision, 60 rue du Carignan, 34270 Valflaunès, France; christophe.montagnon@rd2vision.com

**Keywords:** Nagoya Protocol, Convention on Biological Diversity, coffee industry, genetic resources, benefit-sharing, compliance strategies, international agreements, coffee research, sustainable coffee cultivation, biodiversity conservation

## Abstract

The Nagoya Protocol on Access and Benefit-sharing (ABS) of the Convention on Biological Diversity (CBD) is a fundamental international agreement that plays a crucial role in the protection and equitable utilization of plant genetic resources. While this agreement is essential for conservation and sustainable use, it presents specific challenges to coffee research and industry. One major issue is the requirement to obtain prior informed consent (PIC) from the source country or community, which can be a complex and time-consuming process, especially in regions with limited governance capacity. Additionally, the mandates of this agreement necessitate benefit-sharing with the source community, a requirement that poses implementation challenges, particularly for small businesses or individual researchers. Despite these challenges, the importance of the Nagoya Protocol in the coffee sector cannot be overstated. It contributes significantly to the conservation of coffee genetic resources and the sustainable utilization of these resources, ensuring fair distribution of benefits. To address the complexities presented by this international framework, coffee researchers and industry need to engage proactively with source countries and communities. This includes developing clear and equitable benefit-sharing and implementing strategies for compliance. This article explores the impact of the Nagoya Protocol on the coffee industry, particularly emphasizing the need for balancing scientific investigation with the ethical considerations of resource sharing. It also discusses practical strategies for navigating the complexities of this agreement, including research focused on authenticity control and the challenges in conducting large-scale coffee studies. The conclusion underscores the potential for international collaboration, particularly through platforms like the International Coffee Organization (ICO), to harmonize research activities with the ethical imperatives of the Nagoya Protocol.

## 1. Introduction

The history of the Nagoya Protocol [1], a pivotal international agreement, can be traced back to the late 20th century. It finds its roots in the initiatives of the United Nations Environment Programme (UNEP) which, in 1988, established a working group of Experts on Biological Diversity that laid the groundwork for future developments in environmental conservation and the sustainable use of biodiversity [2].

A landmark moment in this journey occurred on 5 June 1992, during the Earth Summit in Rio de Janeiro. This date marked the opening for signature of the Convention on Biological Diversity (CBD), a comprehensive multinational agreement addressing biological diversity [2]. The significance of 1992 is profound as it established a temporal threshold; actions and acquisitions of biological resources prior to that year are distinguished from those that occurred thereafter under the new international legal framework provided by the CBD.

Following the establishment of the CBD, two critical protocols were developed, each addressing different aspects of biodiversity management. The first was the Cartagena Protocol, adopted in 2000, focusing on biosafety. It emphasizes the safe handling, transfer, and use of living modified organisms resulting from modern biotechnology, ensuring they do not adversely affect biological diversity or human health [3].

The second and equally significant protocol is the Nagoya Protocol, adopted in 2010 (Table 1). This protocol specifically deals with access to genetic resources and the fair and equitable sharing of benefits arising from their utilization (access and benefit-sharing, ABS) [1]. The Nagoya Protocol underscores the importance of recognizing and compensating the contributions of indigenous and local communities who have conserved and sustainably used biodiversity, including the maintenance of traditional knowledge associated with genetic resources. This Protocol partly emerged in response to historical practices, particularly in the medicinal industry, in which commercial entities often exploited natural and indigenous resources without fair compensation. These practices included using local knowledge to identify plants with medicinal properties, commercializing the derived products, and not adequately sharing the benefits with the communities or countries of origin [4]. Interestingly, one of the very active countries in paving the way for the Nagoya protocol was Ethiopia and the motivation of Ethiopian authorities was what Lemma and Maryo [5] named the “monumental biological theft” of Ethiopian national heritage crops, citing explicitly coffee (*Coffea arabica*) as an example [5].

The Nagoya Protocol officially came into force in October 2014, signifying a major step forward in the global commitment to biodiversity conservation and sustainable use [1]. The parties to the Nagoya Protocol, similar to those of the Cartagena Protocol [3] and the CBD [2], are national governments that have ratified the agreement. Information about the member states of these protocols is readily available on the United Nations website, providing transparency and accountability [6].

It is noteworthy that some major countries, like the United States of America (USA), are not parties to the Nagoya Protocol [6], but may still generally support the principles of ABS [7]. For example, the University of California was described to have adopted the Nagoya Protocol as best practice when dealing with biological resources [8].

The coffee sector is facing challenges, including climate change [9,10,11,12,13], which has been well-documented [9,14,15]. Farming systems are already transitioning towards agroforestry [16,17,18,19], but a significant part of the solution will undoubtedly be genetic [20,21,22]. *Coffea arabica* is one of the most genetically narrow cultivated species [23,24], primarily due to drastic bottlenecks during its evolution [25,26]. Furthermore, recent research has revealed that certain genetic populations of the *C. arabica* species, located in South Sudan [27], Yemen [28,29,30], and the Hararghe region of Ethiopia [30], have not been fully utilized due to their previous unknown status. It is important to note that the use of these genetic resources outside of their center of origin may fall under the Nagoya Protocol, as well as any genetic resources that have yet to be explored in Southern Ethiopia. *C. canephora* is a potential replacement for *C. arabica* in regions where the latter species is no longer viable. The genetic diversity of *C. canephora* is vast [31], but only a small portion of it has been studied [32,33,34]. Overcoming specific challenges associated with *C. canephora*, such as cup quality, will require the extensive use of genetic resources found in various West and Central African countries. The Nagoya framework will be significant in this context. Additionally, it is worth noting that *C. arabica* and *C. canephora* are just two of the 130 species in the *Coffea* genus. Some of these species have been identified as potential solutions to the challenges posed by climate change [35,36,37,38,39]. These species, which have potential interest, are found in various African countries and will require significant agronomic research in the future.

Conserving the unique genetic diversity of coffee is crucial for the resilience and sustainability of the global coffee industry, given the challenges it faces. Therefore, it is important for the coffee community to be aware of the Nagoya Protocol and conduct research within its framework.

The aim of this article is not to provide an exhaustive review of all the provisions of the Nagoya Protocol, but rather to describe them sufficiently to raise awareness within the coffee community. The authors propose the creation of a discussion forum within the industry, potentially under the International Coffee Organization, to coordinate research practices and the exchange of coffee genetic resources worldwide.

## 2. Classification of Coffee Genetic Resources under the Nagoya Protocol: Scope and Criteria

The Nagoya Protocol applies a broad definition of ‘genetic resources’. This term encompasses any living tissues or parts of living organisms, indicating a wide-ranging scope of coverage. This definition requires a thorough examination of which genetic resources for coffee are covered by the Protocol.

Primarily, the Nagoya Protocol’s application is contingent on how ‘known’, ‘widespread’, and ‘familiar’ a genetic resource is. For coffee cultivars, many traditional and widespread varieties, especially those of the *C. arabica* species such as Bourbon or Typica, are not covered by the Nagoya Protocol. These varieties have been globally disseminated and utilized for centuries, placing them outside the Protocol’s domain due to their widespread and well-known status. For example, an assessment from Brazil states that Brazil would not have to share the benefits of coffee genetic resources that were introduced into the country before the Protocol came into force [40]. The European Union (EU) has also specifically excluded historical material when implementing the Nagoya Protocol [8].

However, the situation differs for specific wild accessions or even locally domesticated cultivars or landraces, some of which might be found outside of the native habitat of the species. For instance, certain coffee landraces in Timor Leste [41], New Caledonia [42], Yemen [29,30], or in the region of Hararghe in Ethiopia [30], have unique genetic patterns and have been conserved locally over centuries. These landraces are geographically confined and preserved by indigenous communities that the Protocol aims to protect [28,29,30].

In contrast, Ethiopian landraces and wild accessions collected in the 1960s and transferred to different germplasm field collections, such as those in Centro Agronómico Tropical de Investigación y Enseñanza (CATIE) in Costa Rica [43,44], are in theory not covered by the Nagoya Protocol. These resources are considered as freely accessible because the collection occurred before the 1992 cutoff set by the CBD.

It is important to note that adherence to the principles of the Nagoya Protocol is not solely a legal obligation but can also be an ethical choice. For example, institutions like CATIE choose to manage some of their genetic resources as if they were under the Nagoya Protocol, acknowledging their origin and promoting fair benefit-sharing practices, despite not being legally bound to do so (see Section 4) [45].

Regarding coffee, most genetic resources that are not currently cultivated or widely known would most likely fall under the Nagoya Protocol. The case of *C. liberica* illustrates a unique scenario. While Liberica is not a majorly cultivated species and is relatively rare, its historical introduction to Asia [46,47] means that Asian cultivars of Liberica might not typically fall under the Protocol because they were introduced a long time ago, but might be under the protocol if some specific and unique features have been locally selected. However, Liberica genetic resources found in their original African habitats, such as Côte d’Ivoire, Sierra Leone, or the Central African Republic, would likely be subject to Nagoya Protocol regulations due to their localized and lesser-known status.

In conclusion, the application of the Nagoya Protocol to coffee genetic resources hinges on factors like historical dissemination, current cultivation status, and the extent of global knowledge about them. To obtain detailed and specific information about the coffee genetic resources covered by the Nagoya Protocol, it is necessary to consult resources from botanical or agricultural research institutions or to make a direct inquiry with authorities managing the implementation of the Nagoya Protocol.

## 3. Nagoya Protocol for Genetic Resources: Short Background on Access and Benefit-Sharing (ABS)

The inclusion of genetic resources under the Nagoya Protocol has significant implications, particularly in the realms of research and knowledge production because it becomes crucial to adhere to the protocol’s provisions and guidelines, which are geared towards equitable sharing of benefits derived from such resources.

### 3.1. Prior Informed Consent (PIC)

One of the fundamental aspects of working under the Nagoya Protocol is obtaining Prior Informed Consent (PIC) from the government of the country where the genetic resources were originally acquired. Note that each party to the protocol must identify a national focal point which can be found on the website of the protocol [48]. The process of obtaining a PIC involves contacting the relevant government before beginning any research or knowledge production activities. This interaction ensures that the country providing the genetic resources is aware of and consents to their use. In cases where there is no response to a PIC request, it might be considered a positive response, allowing researchers to proceed, provided they have documented their due diligence efforts.

### 3.2. Mutually Agreed Terms (MAT)

Beyond obtaining PIC, the Nagoya Protocol necessitates establishing Mutually Agreed Terms (MAT) between the user and the provider of genetic resources. MAT outlines the specific terms of agreement, including any benefit-sharing arrangements. This agreement is vital as it clarifies the expectations and responsibilities of both parties involved in the exchange and use of genetic resources.

### 3.3. Bilateral Agreements and Bargaining

The Nagoya Protocol operates on a bilateral basis, meaning that agreements and terms can vary between different entities and for different resources. A country can have distinct agreements with different organizations for the same genetic resources, reflecting the bilateral nature of these negotiations.

### 3.4. Due Diligence and Communication with Focal Points

Researchers and organizations must demonstrate due diligence in reaching out to national focal points [48], which are listed on the Nagoya or UN website for each country. This due diligence is critical, even if getting responses from national focal points may be challenging due to varying levels of organization and preparedness across countries.

### 3.5. Distinction from the International Plant Treaty

It is important to distinguish the Nagoya Protocol from the International Treaty on Plant Genetic Resources for Food and Agriculture (ITPRFA), which operates a multilateral system for certain essential food crops deemed crucial for global food security [49]. The International Plant Treaty does not include coffee [45,50], and thus coffee genetic resources fall under the bilateral negotiation framework of the Nagoya Protocol.

### 3.6. Provider’s Role and Community Acknowledgment

Under the Nagoya Protocol, the provider of genetic resources—typically the government representing its people—plays a key role in setting the terms for ABS. It is essential that these agreements not only reflect the government’s stance but also the interests and acknowledgment of the local communities that have conserved these resources. The government acts as a representative between the external entities and its communities to ensure that any agreement under the Nagoya Protocol benefits the community effectively.

### 3.7. Monitoring and Regulatory Compliance under the Nagoya Protocol

The enforcement and control mechanisms of the Nagoya Protocol are critical for its effective implementation, particularly in regulating the movement and use of genetic resources. The control and enforcement of the Nagoya Protocol primarily occurs at the country’s entry points, with the importing country bearing the responsibility for compliance checks. This system places a significant emphasis on the importing entities to provide necessary documentation or evidence of due diligence.

#### 3.7.1. Role of Entry Points in Control

The primary control occurs at the entry points of a country. For instance, if genetic resources are transported from Kenya to France, the French authorities are responsible for checking these resources upon entry. The compliance check includes verifying whether the resources align with the Nagoya Protocol provisions. French authorities would ensure compliance with the Nagoya Protocol, including verification of Prior Informed Consent (PIC) and Mutually Agreed Terms (MAT) with Kenya.

#### 3.7.2. Responsibility of the Importer

The importer is responsible for demonstrating compliance with the Nagoya Protocol. This could involve presenting PIC documentation or evidence of attempts to communicate with the country of origin. If the importing individual or entity has fulfilled their due diligence but received no response, this is taken into account by the entry point authorities.

#### 3.7.3. Post-Entry Controls and Reporting Mechanisms

Besides entry point checks, there are mechanisms for post-entry oversight. These could be random checks or investigations triggered by reports from external sources. For instance, if a country suspects non-compliance or deceit in the use of its genetic resources in another country, it can request the corresponding authorities for a re-evaluation. Such mechanisms ensure ongoing surveillance and compliance even after the genetic resources have entered a country.

#### 3.7.4. Mechanisms for Nagoya Protocol Disputes

The Nagoya Protocol itself does not establish a specific international court for resolving trade disputes related to its provisions (note: this is not explicitly stated in the Nagoya Protocol but is based on the absence of such a provision in the document). The Nagoya Protocol encourages parties to resolve disputes through negotiation and mediation (Article 18 [1]). However, it does not specify a particular international mechanism for such resolution, relying instead on parties to find amicable solutions. If disputes cannot be resolved through negotiation or mediation, the Nagoya Protocol suggests considering arbitration or other peaceful means (Article 18 [1]). This article encourages parties to use mechanisms like arbitration but does not mandate a specific international forum. In practice, disputes might be addressed through national courts in the jurisdiction where the alleged breach occurred or through international arbitration if agreed upon by the parties involved. This approach is based on common practice in international law, as the Nagoya Protocol does not provide detailed procedures for such dispute resolution mechanisms. Each Party to the Nagoya Protocol is required to designate one or more national competent authorities to handle ABS matters (Article 13 [1]). These authorities may also play a role in dispute resolution within their jurisdictions, although the specific nature of this role is not detailed in the Protocol and is based on common practice in international law. For international matters involving trade disputes, mechanisms like the World Trade Organization (WTO) might be relevant, based on common practice in international law and the general principles of the WTO’s involvement in international trade disputes, as the Nagoya Protocol does not directly reference the WTO.

In summary, while the Nagoya Protocol (particularly Article 18) acknowledges the possibility of disputes and suggests amicable resolution methods, it does not establish a specific international court for this purpose. The practical resolution of disputes under the Protocol is largely guided by common practices in international law and the legal frameworks of the countries involved.

## 4. Approaches to Nagoya Protocol Implementation in the Coffee Landscape

Coffee-producing countries that have not ratified the protocol include Colombia, Costa Rica, El Salvador, Haiti, Jamaica, Papua New Guinea, Thailand, Timor-Leste, and Yemen [6] (note that some countries such as Colombia, El Salvador, Costa Rica, Thailand, and Yemen have signed but not ratified the protocol, which is equivalent to no adoption). This absence raises various questions and considerations in the context of international biodiversity management and benefit-sharing. Nonetheless, the members and parties to these protocols represent a global consensus on the importance of preserving biodiversity, ensuring biosafety, and promoting equitable benefit-sharing, principles crucial for the well-being of the planet and humankind.

The implementation of the Nagoya Protocol in the coffee sector reflects a commitment to ethical practices and acknowledgment of source communities for genetic resources, even by non-state actors, such as universities, herbaria, and scientific journals, who have made Nagoya compliance a pre-condition of dealing with biological materials [8]. This is specifically evident in the approaches of various coffee research institutions, including the *Coffea* Biological Resource Centre (BRC) in La Réunion, France, CATIE in Costa Rica, and the International conservation collection of Coffee varieties at the Zoological and Botanical Garden Wilhelma in Stuttgart, Germany [51,52].

The *Coffea* BRC, managed jointly by the French Institut de Recherche Pour Le Développement (IRD) and Centre de Coopération Internationale en Recherche Agronomique pour le Développement (CIRAD) [53], is dedicated to preserving wild coffee species. It holds over 700 genotypes from various African countries and Indian Ocean islands, stored in a seed vault in Montpellier and a field collection in southern Réunion Island. Despite many genetic resources being acquired from the 1960s to the 1980s before the Nagoya Protocol’s enforcement [54], the BRC adheres to its principles voluntarily. This includes informing researchers about the origins of these resources and aligning exchanges with the Nagoya Protocol’s guidelines on traceability and benefit-sharing.

CATIE in Costa Rica has similarly chosen to act in accordance with the Nagoya Protocol for many of its genetic resources. This proactive approach ensures that external parties accessing these resources are informed of their origin and that the terms align with the Protocol’s principles, irrespective of legal obligations.

The approach of the Zoological and Botanical Garden Wilhelma in Stuttgart also exemplifies strict compliance with the Nagoya Protocol. The Wilhelma has implemented rigorous guidelines, supported by the German Federal Office for Agriculture and Food, to ensure that research on coffee varieties is conducted within the Protocol’s framework [51]. This approach reflects a cautious and responsible attitude towards the use of genetic resources, ensuring adherence to ethical and legal standards.

These institutions, through their varied but dedicated approaches, highlight the evolving understanding and implementation of the Nagoya Protocol in the coffee sector. The commitment to ethical research practices and recognition of the importance of benefit-sharing and acknowledging the source communities for genetic resources underscores the role of these institutions in promoting responsible use and conservation of coffee genetic resources.

## 5. The Impact of the Nagoya Protocol on Global Coffee Studies

The Nagoya Protocol can pose significant challenges for conducting expansive research studies, especially in the field of coffee. This is particularly evident when considering the logistical and bureaucratic complexities associated with obtaining necessary approvals from various countries. An example of the few currently available scientific studies on coffee made in compliance with Nagoya Protocol is Raharimalala et al., which studied *C. humblotiana*, a wild species from the Union of the Comoros [55]. The certificate for compliance can be checked at the ABS clearing-house website [56].

In cases where research involves analyzing coffee samples from multiple countries, researchers may face the daunting task of securing approvals from a large number of governments. For instance, a study encompassing coffee from around the globe could require contacting and obtaining consent from upwards of fifty different national authorities. This bureaucratic process alone can be a significant deterrent, potentially discouraging researchers from undertaking such comprehensive studies.

Another critical factor influencing the feasibility of research under the Nagoya Protocol is the date of sample collection. If coffee samples were collected before 1992, they would not technically fall under the Protocol’s requirements. However, many institutions and authorities opt to treat all samples, regardless of their collection date, as if they were obtained post-1992. This approach aims to maintain consistency and uphold the ethical standards of the Protocol but can inadvertently create additional barriers for researchers.

Institutions like CIRAD or Zoological and Botanical Garden Wilhelma (see Section 4) exemplify a rigorous approach to Nagoya compliance, emphasizing the importance of adhering to the Protocol’s guidelines even if it means additional time and effort for researchers. While this approach is not intended to be obstructive, the practical effect can be akin to a research blockage. The complexities and time-consuming nature of obtaining prior informed consent from multiple countries can be so overwhelming that researchers might opt to redirect their efforts to other areas where such hurdles are absent.

The Nagoya Protocol’s impact on coffee research highlights a critical balance that needs to be struck between ethical obligations to source countries and the practicalities of conducting large-scale, multinational research. While the Protocol serves a crucial role in protecting genetic resources and the rights of source communities, its implementation can inadvertently impede scientific exploration and discovery, particularly in studies requiring a broad and diverse range of samples.

From a researcher’s perspective, although the Nagoya Protocol may sometimes pose obstacles or cause frustration due to its stringent requirements and bureaucratic processes, these challenges are viewed as necessary. The Protocol introduces a level of difficulty that is deemed essential for safeguarding the long-term interests of biodiversity conservation and equitable benefit-sharing. The rigorous nature of the Nagoya Protocol’s compliance mechanisms ensures that research on genetic resources is conducted ethically and responsibly.

## 6. Evaluating the Nagoya Protocol in the Coffee Sector: Balancing Research Challenges with Ethical Equity

The Nagoya Protocol, from the perspective of both a citizen and a researcher, presents a complex yet necessary landscape in the realm of biodiversity and genetic resource management. This duality is particularly evident in the context of coffee research and the implications of the Protocol on fair and equitable benefit-sharing.

As a citizen, the Nagoya Protocol is seen as a very positive process. It addresses historical inequities in the utilization of genetic resources and ensures that the benefits derived from such resources are shared fairly. This aspect of the Protocol is crucial for correcting past injustices and promoting a more equitable distribution of resources and benefits, especially for countries that have historically provided these resources without receiving adequate recognition or compensation. Had the Nagoya Protocol been in place in the 1990s and 2000s for resources introduced from origins during the 20th century, who knows what would have been the story of Geisha, which is sold up to USD 350 a pound [57], and if and how Ethiopian communities that have maintained these resources on site would have been acknowledged or rewarded? The Nagoya Protocol aims to address such situations, ensuring that countries like Ethiopia, which have contributed valuable genetic resources, are duly recognized and eventually compensated. Implementing specialized ABS mechanisms under the Nagoya Protocol could significantly aid in preserving Arabica coffee’s genetic resources. By enforcing strict land-use zoning and conservation strategies, these mechanisms promise to generate substantial funds from the global coffee industry. This financial support is vital for the long-term conservation of coffee genetic resources, ensuring their sustainable use while maintaining fair access to these genetic resources [43,58].

## 7. Conclusions

Addressing the complexities of the Nagoya Protocol in the coffee industry and research involves multiple strategic approaches, foremost among them being the establishment of clear guidelines and a comprehensive list of coffee genetic resources. This list would distinguish between genetic resources covered by the Nagoya Protocol and those exempt, providing researchers and stakeholders with essential clarity and simplifying compliance procedures. Such a categorization could significantly streamline research processes, allowing for more focused studies on coffee genetics and breeding, while respecting the ethical principles of equitable resource sharing. Another option would be to add coffee into the multilateral system of ITPRFA [8].

Ensuring the authorities’ ability to verify coffee upon its entry is hindered by restricted access to authentic specimens due to the Protocol’s regulations. Therefore, creating specific exemptions for such control research is vital to enable effective and efficient monitoring of coffee in the market without being impeded by regulatory barriers. Clarification could also be made during revision of the Nagoya Protocol regarding how to deal with historical plant collections [8].

In parallel, developing a consensus on uniform guidelines for coffee research under the Nagoya Protocol could be beneficial. These guidelines would help harmonize practices across different countries, ensuring a consistent approach to benefit-sharing and genetic resource utilization. This standardization could mitigate some of the bureaucratic challenges currently faced by researchers, fostering more effective international collaboration.

As part of these efforts, engaging with an international platform like the International Coffee Organization (ICO) [59] could be a pivotal step. The ICO, encompassing both consumer and producer governments [60], offers a broad and inclusive forum where these proposed guidelines and lists could be discussed, refined, and potentially adopted. The ICO’s global reach and influence make it a suitable platform for advocating these changes and for seeking broader agreement among the international community. Through such collaborative and multi-faceted efforts, the goal of balancing the facilitation of coffee research with the ethical imperatives of the Nagoya Protocol can be more feasibly achieved.

In conclusion, the Nagoya Protocol plays an increasingly pivotal role as the coffee industry confronts the escalating challenges of climate change, such as prolonged droughts, diseases, and pest infestations [54,61]. Accessions of wild *C. arabica, C. canephora*, *C. liberica* and possibly also other species, under the protective umbrella of this protocol, are poised to be invaluable resources for the genetic enhancement of these cultivated species. Urgent action is clearly needed to conserve the unique, remaining genetic diversity of *C. arabica* in Ethiopia, Yemen, and Sudan [27,29,58]. The implementation of this international framework not only safeguards these vital genetic resources but also ensures their responsible and equitable use, contributing significantly to the resilience and sustainability of the global coffee industry in the face of environmental adversities.

## Figures and Tables

**Table 1 foods-13-00254-t001:** Chronological development of the Nagoya Protocol and preceding biodiversity agreements.

Year	Milestone	Contribution
1988	Establishment of United Nations Environment Programme (UNEP) Working Group on Biological Diversity	Began exploration of global biodiversity issues under UNEP.
1989	Technical and Legal Experts Group formed	Prepared for an international legal instrument on biodiversity conservation and sustainable use.
1991	Intergovernmental Negotiating Committee convened	Negotiated the text of the Convention on Biological Diversity (CBD).
May 1992	Agreed Text of the CBD adopted in Nairobi	Finalized the CBD text.
June 1992	CBD opened for signature at Rio Earth Summit	Marked the global commitment to biodiversity conservation.
December 1993	CBD enters into force	Legal enactment of the CBD.
2001	Adoption of the International Treaty on Plant Genetic Resources for Food and Agriculture (ITPGRFA) at FAO Conference	Addressed plant genetic resources for food and agriculture.
2003	Cartagena Protocol on Biosafety adopted	Focused on biosafety in the handling of modified organisms.
2010	Nagoya Protocol adopted	Established legal framework for access and benefit-sharing (ABS) of genetic resources.
October 2014	Nagoya Protocol enters into force	Began the operational phase of the Nagoya Protocol.

## Data Availability

No new data were created or analyzed in this study. Data sharing is not applicable to this article.

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
