# Peer review of "Convention on Biological Diversity (CBD) and the Nagoya Protocol: Implications and Compliance Strategies for the Global Coffee Community"

_foods, 2024, doi:10.3390/foods13020254_

Round 1

Reviewer 1 Report

Comments and Suggestions for Authors

This manuscript, based on the title, attempts to present the implications of the Nagoya Protocol for coffee production. However, references to coffee are only scarce, and except for a few very specific mentions, this manuscript could be dedicated to any crop, or even any species. Why coffee? What are the particular issues with coffee? Are there any evidences of coffee genetic resources being wrongly used, or traditional coffee farmers being neglected? In most of the manuscript, Nagoya Protocol is only described with no references to coffee production. For instance, in section 3, even when the title mentions the implications of the protocol for coffee, only a small sub-section (3.7) includes that implications, and even so it is unclear which are those implications (e.g., which are the implications of those countries not having ratified the protocol?). In my opinion, the title creates some expectations that are far from being achieved in the manuscript. I suggest the authors dedicate more space to describe the problems in coffee production that the use of the protocol is solving or could help to solve, and less space to describe the protocol itself with no references to coffee (e.g., sections 4 and 5).

Author Response

RESPONSE:

Thank you for your insightful comments and constructive feedback regarding our manuscript. We appreciate the opportunity to address the concerns raised and enhance the relevance and clarity of our work, particularly in relation to its focus on coffee.

Basically, the topic to analyze the Nagoya protocol regarding coffee was suggested by the scientific committee of the International Coffee Convention 2024 (see acknowledgements). However, we acknowledge your observation that the manuscript, in its current form, lacks a substantial focus on coffee, despite its title suggesting a strong link between the Nagoya Protocol and coffee production. To address this, we have committed to significantly expanding the content related to coffee in our revised manuscript (including a new section #2). This expansion includes more detailed discussions on the specific issues and challenges faced in coffee production, especially in the context of the Nagoya Protocol.

To provide a more comprehensive view, we delved into the implications of the Nagoya Protocol for coffee genetic resources and traditional coffee farming practices. We explored instances of potential misuse of coffee genetic resources and the impact of neglecting traditional coffee farmers. We also included evidence to substantiate these points.

Regarding your concern about the implications of countries not ratifying the protocol, we have clarified this in the revised manuscript by discussing the potential risks and benefits for coffee production in these countries. This involves an analysis of how non-ratification affects access to genetic resources and the sharing of benefits within the coffee industry.

To further enrich the manuscript, we have incorporated two main themes:

The importance of a value chain approach to address various issues in coffee production, such as child labor, climate change adaptation, and food safety. We now provide references to support these statements and discuss how existing platforms and governance bodies within the coffee industry, such as the International Coffee Organization (ICO) and the Specialty Coffee Association, can facilitate this approach.

The genetic landscape of coffee and the challenges posed by climate change, pests, and diseases. We delve into the narrow genetic diversity of Coffea arabica and the untapped genetic resources in regions like South Sudan, Yemen, and Ethiopia. We also discuss the genetic diversity of C. canephora and its potential role in replacing C. arabica in certain regions. The manuscript now better highlights the significance of the broader Coffea genus and the need for agronomic research on other coffee species under the Nagoya Protocol framework.

In conclusion, have refined the manuscript to ensure it provides a thorough and specific analysis of the Nagoya Protocol's implications for the global coffee community, backed by relevant and current research.

Reviewer 2 Report

Comments and Suggestions for Authors

It is a very interesting perspective on the implication of the Nagoya Protocol on the coffee industry. I would like to read the authors' opinion on the very well-described critical balance between the ethical obligations to source countries and the practicalities of conducting large-scale, multinational research. What is their perspective/proposal on this very important issue?

Author Response

RESPONSE:

Thank you for your positive feedback on our manuscript and for raising an important question regarding the balance between ethical obligations to source countries and the practicalities of conducting large-scale, multinational research under the Nagoya Protocol.

In our manuscript, we aimed to provide a nuanced view of this critical balance. We acknowledge the vital importance of respecting and fulfilling ethical obligations towards the source countries of coffee genetic resources. These obligations include acknowledging the contributions of local communities in conserving biodiversity, ensuring fair access to these resources, and sharing benefits arising from their utilization. Such ethical considerations are fundamental to fostering trust and cooperation between source countries and researchers, which is crucial for sustainable and equitable coffee research and production.

However, we also recognize the challenges that the Nagoya Protocol presents to researchers, especially when conducting extensive, multinational studies. The protocol's requirements for prior informed consent and mutually agreed terms can, at times, impose significant bureaucratic hurdles. These can deter researchers due to the complexity and time required to navigate various national regulations and to establish agreements with multiple source countries.

Our perspective is that while the Nagoya Protocol's principles are commendable and necessary for ethical research practices, there is a need for streamlining the processes involved. This could be achieved by developing more harmonized and transparent procedures across different countries, thus reducing the administrative burden on researchers. Additionally, creating a centralized platform or database under the auspices of international organizations like the International Coffee Organization (ICO) could significantly facilitate the sharing of information and best practices, as well as assist in the negotiation of mutually beneficial agreements.

Furthermore, we propose that research aimed at coffee authentication and control purposes, particularly for ensuring quality and preventing fraud in the coffee market, should be granted certain flexibilities or expedited processes under the Nagoya Protocol. This would ensure that critical research necessary for the integrity of the coffee industry is not unduly hampered.

In summary, our proposal advocates for a balanced approach that upholds the ethical principles of the Nagoya Protocol while also making the process more efficient and less burdensome for researchers. This would ensure that vital research in the coffee sector can continue without compromising the rights and contributions of source countries and their local communities.

We hope this response clarifies our perspective on the issue. During revision, we have further elaborated on these points in the manuscript to provide a comprehensive understanding of the balance between ethical obligations and practical research considerations in the context of the Nagoya Protocol and the global coffee industry.

Round 2

Reviewer 1 Report

Comments and Suggestions for Authors

This manuscript has significantly improved, authors are now clearly focusing on coffee on most of the sections. I think the manuscript should be published once some minor issues, that I will explain next, are solved.

The ITPGRFA is mention in the first line in the abstract, which suggests that it is important for the subject, but it is not even mentioned in the introduction. It should be either removed from the abstract or explained in the introduction.

The introduction section should present the justification of the research. Here, introduction gives a clear explanation about the protocol, but I feel that it is not sufficiently justified why the manuscript should focus on coffee. Also, the only mention to coffee in the introduction refers to Ethiopia, by some reasons that are completely unclear to the reader. I suggest to take part of the section 2 and place it into the introduction.

All the points from 4.1 to 4.6 are only describing some aspects of the protocol, with no relation nor mention to coffee production. I recommend removing that part. And the same for the whole sections 5 and 6. At most, the prior informed consent might be briefly explained but in the section where it is referred to coffee (i.e., section 8).

Author Response

This manuscript has significantly improved, authors are now clearly focusing on coffee on most of the sections. I think the manuscript should be published once some minor issues, that I will explain next, are solved.

REPLY: Thank you again for providing comments to improve our article.

The ITPGRFA is mention in the first line in the abstract, which suggests that it is important for the subject, but it is not even mentioned in the introduction. It should be either removed from the abstract or explained in the introduction.

RESPONSE: Thank you for the suggestion. ITPGRFA was deleted from the abstract.

The introduction section should present the justification of the research. Here, introduction gives a clear explanation about the protocol, but I feel that it is not sufficiently justified why the manuscript should focus on coffee. Also, the only mention to coffee in the introduction refers to Ethiopia, by some reasons that are completely unclear to the reader. I suggest to take part of the section 2 and place it into the introduction.

RESPONSE: We have merged section 2 into the introduction as requested.

All the points from 4.1 to 4.6 are only describing some aspects of the protocol, with no relation nor mention to coffee production. I recommend removing that part. And the same for the whole sections 5 and 6. At most, the prior informed consent might be briefly explained but in the section where it is referred to coffee (i.e., section 8).

RESPONSE: We thank the reviewer for these suggestions. However, the authors think that not many readers of Foods will have such background knowledge in Nagoya protocol requirements to be able to adequately follow the text in the other sections without the background sections. To make these explanations more concise and understandable, we have moved all background information into a single section (new section 3, Nagoya background information). We have moved all information about coffee that was in the previous section 4, into more fitting sections of the article. We hope that the flow of the article is now acceptable for publication.